# Characterization of Conductive Carbon Nanotubes/Polymer Composites for Stretchable Sensors and Transducers

**DOI:** 10.3390/molecules28041764

**Published:** 2023-02-13

**Authors:** Laura Fazi, Carla Andreani, Cadia D’Ottavi, Leonardo Duranti, Pietro Morales, Enrico Preziosi, Anna Prioriello, Giovanni Romanelli, Valerio Scacco, Roberto Senesi, Silvia Licoccia

**Affiliations:** 1NAST Centre, University of Rome Tor Vergata, 00133 Rome, Italy; 2Department of Chemical Science and Technologies, University of Rome Tor Vergata, 00133 Rome, Italy; 3Department of Physics, University of Rome Tor Vergata, 00133 Rome, Italy; 4School of Neutron Spectroscopy SONS, University of Rome Tor Vergata, 00133 Rome, Italy

**Keywords:** carbon nanotubes, polymer composites, self-assembly, stretchable sensors, stretchable conductors

## Abstract

The increasing interest in stretchable conductive composite materials, that can be versatile and suitable for wide-ranging application, has sparked a growing demand for studies of scalable fabrication techniques and specifically tailored geometries. Thanks to the combination of the conductivity and robustness of carbon nanotube (CNT) materials with the viscoelastic properties of polymer films, in particular their stretchability, “surface composites” made of a CNT on polymeric films are a promising way to obtain a low-cost, conductive, elastic, moldable, and patternable material. The use of polymers selected for specific applications, however, requires targeted studies to deeply understand the interface interactions between a CNT and the surface of such polymer films, and in particular the stability and durability of a CNT grafting onto the polymer itself. Here, we present an investigation of the interface properties for a selected group of polymer film substrates with different viscoelastic properties by means of a series of different and complementary experimental techniques. Specifically, we studied the interaction of a single-wall carbon nanotube (SWCNT) deposited on two couples of different polymeric substrates, each one chosen as representative of thermoplastic polymers (i.e., low-density polyethylene (LDPE) and polypropylene (PP)) and thermosetting elastomers (i.e., polyisoprene (PI) and polydimethylsiloxane (PDMS)), respectively. Our results demonstrate that the characteristics of the interface significantly differ for the two classes of polymers with a deeper penetration (up to about 100 μm) into the polymer bulk for the thermosetting substrates. Consequently, the resistance per unit length varies in different ranges, from 1–10 kΩ/cm for typical thermoplastic composite devices (30
μm thick and 2 mm wide) to 0.5–3 MΩ/cm for typical thermosetting elastomer devices (150 μm thick and 2 mm wide). For these reasons, the composites show the different mechanical and electrical responses, therefore suggesting different areas of application of the devices based on such materials.

## 1. Introduction

The recent advances in the development of composite materials aim at achieving stretchable and flexible materials endowed with tunable electrical properties [1,2,3,4,5,6,7,8,9]. The development of electromechanical sensors based on such materials, through simple and low-cost procedures, is paramount to optimize manufacturing processes. The process of upgrading to the large-scale production of electromechanical sensors implies in particular the possibility of a wide-ranging extension to biomedical application, aiming at the easy and constant monitoring of patients, and consequently at the improvement in their quality of life. Electrical conductivity varies with the concentration of the conductive component, and it has been reported that it can be made more or less dependent on the strain to fabricate strain sensors with different sensitivities [7,10,11]. Many computational and experimental techniques have been used to study these composites, ranging from numerical simulations to several experimental techniques, the most widely used being scanning electron microscopy (SEM) [12,13,14] for a surface analysis and confocal Raman microscopy measurements [15,16,17,18] for a bulk analysis. The fabrication procedures and subsequent processing have also been reported to significantly affect the composite electromechanical properties [19]. We have previously developed single-wall carbon nanotube (SWCNT, also indicated simply as CNT in the following) polyethylene composite films, combining the thermoplastic properties of the polymer and the conductive and elastic properties of the SWCNT bundles. We have obtained arrays of submillimetric conductive tracks on polyethylene films that have been stretched, shaped, and implanted on the brain cortex of laboratory rats, allowing, for the first time, to successfully probe their electro-corticographic signals for several months [19,20,21].

In the present work, we have extended the previous study analyzing, by additional techniques, the SWCNT interface with two thermoplastic polymers (low-density polyethylene (LDPE) and polypropylene (PP)) and two thermosetting elastomers (polyisoprene (PI) and polydimethylsiloxane (PDMS)) to gain a deeper understanding of the characteristics of the interwoven polymer chains and carbon nanotube bundles.

We have therefore characterized the SWCNT self-grafting on the surface and the gradient of their penetration inside the polymer films, comparing the two polymeric classes. In addition, we have investigated the consequences of the two different grafting mechanisms on the mechanical and electrical behavior of the prepared composites, with the aim of developing different types of electrical sensing devices for application in medicine and prosthetics as well as in other engineering fields.

## 2. Results

To have a first insight into the type of grafting of CNTs onto the different polymeric substrates, the first characterization of the self-grafted SWCNT layer was obtained by the optical profilometry of the edge between a conductive track made of CNT bundles deposited by the drop casting of an aqueous suspension of SWCNTs and the polymeric substrates. Figure 1 shows the SWCNT tracks on (a) an LDPE and (b) PP films having a thickness of around about 25 μm (LDPE) and 35 μm (PP). A compact track and a sharp step can be observed for both thermoplastic polymers. Table 1 shows the average step height compared to the root mean square (RMS) roughness of the polymeric substrates after the manual brushing and cleaning performed to remove nonadhering CNT flakes (as described in Materials and Methods). The polished sample obtained by depositing the conducting tracks on polyisoprene shows a negative height due to the removal of the superficial polymer layer together with the surface-grafted CNT.

Images from the optical profilometer complement the previous investigation obtained by scanning electron microscopy (SEM): the tracks, 3 × 30 mm in size, have thickness values which can be varied by increasing the number of drop depositions (2–5 μm per single deposited drop of the nanotube suspension) [19,20]. The SWCNT-deposited layer remains grafted onto the polymer surface, even after the manual brushing performed to remove the residual non-grafted SWCNT flakes. The resulting conductive track sticking onto the polymeric film, stabilized by the penetration of the viscous polymer by capillary forces [19], has a relatively low electrical resistance per unit length, of the order of 1–10 kΩ/cm.

A completely different behavior is observed by optical profilometry for SWCNT tracks deposited on thermosetting elastomers, reported in Figure 1 for the PI (c) and PDMS (d). For this type of polymer, polishing the tracks by mechanical brushing leads to an almost complete removal of the bundles deposited above the surface, but nanotube bundles appear to be anchored deeply into the polymer structure, remaining buried in the polymer bulk. The resulting conductive track, grafted and immersed in the elastomer film, shows a resistance per unit length of the order of 0.5–3 MΩ/cm (about two orders of magnitude higher than on the thermoplastic substrates). Because of the non-uniform volume distribution of the conductive component into the polymer, the electrical characteristics are reported as the resistance per unit length of the conductor rather than the material resistivity.

Figure 2 shows the SEM micrographs of the cross sections of the four samples at different magnifications chosen to highlight the specific characteristics of each sample. The images confirm and explain this different behavior by showing the following:For the thermoplastic materials (Figure 2a,b), a fairly well-defined layer of the CNT, where single bundles are easily observed, covers the polymer surface. Looking at the details of the images, one can note the polymer soaking up the CNT layer and accumulating at its surface, thus stabilizing the deposition (see also Section A.1 for detail).For the elastomeric substrates (Figure 2c,d), the CNT bundles are hardly ever visible in the film section, being immersed in the polymer matrix, and rarely show up as individual “ropes” coated by the polymer, protruding out of the film section.

The penetration gradient of the CNT inside the different polymers was investigated by confocal Raman microscopy. Such a technique allows high-resolution chemical imaging of the samples by a combination of the spectral information acquired through Raman spectroscopy and of the spatial filtering associated with the confocal optical microscope. The Raman spectral bands derive from the characteristic vibrational modes of the molecules within the samples. When the bands related to these modes are well separated, proper regions of the spectra can be selected to obtain the chemical-physical information on the single components of the materials under investigation. The confocal optics of the microscope allows a volume analysis of the samples by collecting a series of Raman spectra, both on the plane (X, Y) and along the vertical axes (Z). Such a technique can be thus used to obtain the tridimensional information of the composition through the samples. We have thus obtained 3D mapping of the Raman spectroscopic signature of the different materials while scanning the focused laser beam on the plane parallel to the surface of the polymer, for many different planes along its thickness. Carbon nanotubes do not show any significant contribution in the region where the Raman C-H stretching modes due to the polymers appear (2850–2950 cm${}^{-1}$). Such peaks have thus been selected to identify the contribution of the polymeric substrates to the Raman maps. The region related to the Raman G-band (1580–1600 cm${}^{-1}$) was selected to identify the contribution from the CNT in the 3D maps (the details of such spectral signatures are reported in Section A.2).

Figure 3 reports such maps for the different CNT/polymer composites. These maps were acquired by scanning the laser from the pristine polymer face to the CNT-coated side, along planes parallel to the film surface. Such an acquisition procedure allows the laser penetration through the specimen, while otherwise the laser would be absorbed by the CNT layer, without reaching the clean polymer region. Such maps provide an estimate of the maximum depth of the penetration of the SWCNT deposition (the Z coordinate of the plane which first displays a scattering signal from the G-band of SWCNT). As explained in the Materials and Methods section, bright red (green) regions are related to a high CNT (polymer) concentration, as the integral of the Raman contributions related to the CNT (polymer) is high. Yellow and orange regions occur where there is a superposition of both the CNT and polymer contributions (red + green = yellow in the RGB color scheme).

Figure 3a shows the 3D map of a 25 μm thick LDPE film on which the SWCNT was deposited. Nanotubes appear to penetrate up to about 15 μm. This can be assessed from the yellow region that indicates the superposition of the CNT and polymer signals. Figure 3b shows that the SWCNT on a 35 μm thick PP film has a similar behavior, with an average maximum penetration depth of about 25 μm.

Figure 3c refers to a 120 μm thick PI film that, on the contrary, displays a very extended yellow region that proves that CNT bundles penetrate up to about 100 μm. Figure 3d, related to a 250 μm thick PDMS film substrate, shows an SWCNT penetration of up to about 70–80 μm, similar to the case of the PI substrates. Because the laser beam is absorbed by the first layers of the CNT bundles, it was not possible to reconstruct the Raman map along the overall thickness of the specimen, leaving out an unexplored “dark volume”.

Two representative electromechanical characterizations for the CNT conductive tracks on the thermoplastic (LDPE) and thermosetting elastomer (PDMS) are reported in Figure 4. They show very different responses: the electrical resistance of the tracks obtained by the same procedure is highly different for the thermoplastic and elastomeric substrates.

Although the electrical resistance of the nanotube bundle tracks deposited on the thermoplastics is fairly low, such resistance increases rapidly, even over the very limited range of the strains explored (up to 20%). Furthermore, such changes in both the resistance and stress are irreversible: as one can see from cycling back the strain to zero, the resistance does not get back to the original value. On the other hand, the resistance values of the CNT tracks on the elastomeric substrates are higher, their relative increase is much slower, the hysteresis of the strain–release cycle is fairly small and the behavior is almost completely reversible.

The electrical behavior of the different composites appears to be related to their mechanical performance, as shown by the simultaneous recording of the stress–strain and resistance–strain curves on the same specimen. Both types of polymers display hysteresis in both the electrical resistance and the stress due to the relaxation time that the composite structure needs to return to the original configuration once the strain is released. Both hystereses are much larger for the thermoplastic LDPE with respect to the elastomeric PDMS. Possibly, for the 20% strain applied to the LDPE, a permanent plastic deformation of the conductor occurs, associated with a permanent loss of the electrical conductance.

## 3. Materials and Methods

Self-grafted composites were prepared following the previously reported procedure. Polyester stencils were used to produce arrays of wires and electrodes [19,22]. Purified SWCNT, produced by the HiPCo method [23], were purchased from Carbon Nanotechnologies Inc. The diameter of the individual SWCNT is 0.7 nm, with random chiralities from armchair to zig-zag configuration. SWCNT were grafted on different polymer films by the drop-casting technique. A liquid vector solution was prepared by adding 0.1 mg of Linear Alkylbenzene Sulphonate (LAS) to 10 mg of 10:90 wt.% water/ethanol mixture; 1 mg of SWCNT was then added and dispersed in this solution by combination of the surfactant action of LAS with high-power sonication produced by a Fisherbrand model 120 ultrasonic disintegrator for 60 min. Even after such treatment, the SWCNT dispersion was never completely uniform nor stable for a prolonged period of time. However, by keeping the suspension under continuous sonication in a ultrasonic bath (Elmasonic P) operated at 80 kHz, its homogeneity was maintained high enough to flow through a 200 μm needle for the drop-casting purpose. The vector fluid carrying the SWCNT dispersion was deposited by drop casting on polymeric stencils suitably engineered to define the shape of electrical conductors on polymeric films. The polymeric film substrates were supported and fixed onto a metal plate which was heated to a controlled temperature. While the temperature was kept at approximately 80 °C, the CNT suspension was slowly cast through the needle on the stencil. Evaporation of the water/ethanol solution left a brown/black deposition on the polymer surface. The operation was repeated until the empty spaces within the stencils were completely filled and appeared uniformly black. Irrespective of the hydrophobicity of the polymer surface, the vector droplets drying on the hot polymer caused some inevitable reaggregation of nanotube bundles into small flakes of random geometries. Each CNT track was therefore lightly brushed to remove nonadhering flakes and further cleaned by a jet of purified air. To stabilize the deposition, the temperature was raised to a value only slightly lower than the thermoplastic polymer melting temperature (e.g., 115 °C for our LDPE film, 150 °C for PP films). Samples based on thermosetting elastomers (PDMS and PI) were also heated to 150 °C. All heated samples were maintained at the selected temperature for one minute, then cooled to room temperature to undergo a further step of mechanical brushing. Finally, the stencil was carefully removed and the composite devices thus obtained were used for further characterization.

Surface profilometry measurements were carried out using a KLA Zeta-20 optical profilometer equipped with a 50× objective and a focal step (0.04) μm. The optical scanning and step height measurements have allowed extraction of surface profiles and their roughness; the latter was averaged over 5 profiles collected on representative 100 μm^2^
areas from both the pristine polymer and the SWCNT coated portion to assess the thickness of the SWCNT coating and the differences in surface roughness, reported in Table 1. The approximate average thickness of polymer films was measured by a Palmer micrometer, while the optical profilometer supplies more local information on step profiles and roughness, also allowing for 3D surface reconstruction.

SEM scans were obtained by a Tescan Vega (4th Series) electron microscope equipped with a GMU chamber. Secondary electron images were collected in high-vacuum conditions and with beam energies in the 10–20 keV range, selecting the best trade-off in terms of electron penetration and image spatial resolution. To properly study the interface between SWCNTs and the polymers by SEM, the composite films were cut in liquid nitrogen. To obtain a clean section, the samples were soaked in liquid nitrogen to stiffen them, and then they were fractured by means of sharp lancet. Prior to SEM scans, because of their non-conductive nature, the composites were sputter coated with a gold layer of about 10 nm to prevent artifacts due to charge accumulation. The samples were observed by tilting the SEM stage up to 70° angles to find the best position to observe the sides and the edges of the interface layer between the CNT and the polymers, obtaining images as reported in Figure 2.

Confocal Raman microscopy measurements, performed with a Horiba Xplora Plus equipped with a 100 × objective and 638 nm laser excitation, were carried out cutting 2 cm × 2 cm squares from composites that were placed under the microscope, with the naked polymer side facing the objective and the SWCNT layer pointing downward, to reconstruct the tridimensional maps starting from the transparent polymer face. The 3D map was composed of 5 × 5 scanning points (spaced 5 μm) on each of several planes parallel to the surface, corresponding to a number of focal depths (along the Z-axis) with a range depending on the thickness of the polymer. For LDPE, PP, PI and PDMS, the focal depth ranges were 35, 35, 125 and 195 μm, respectively. The z-step for focal distance was approximately 8–10 μm for each composite, with a number of steps depending on the ratio between the depth ranges and the z-step. From the Raman spectra of the composites, two different regions were selected to identify the contributions from SWCNTs and the polymers (see Figure A2). More specifically, to identify the contributions related to SWCNTs, the 1580–1600 cm${}^{-1}$ region around the carbon nanotubes Raman G-band (at about 1591 cm${}^{-1}$) was selected, while for the polymers, the C-H Raman stretching mode (around 2850–2950 cm${}^{-1}$) was selected (for further details, see Section A.2). The 3D maps in Figure 3 were obtained by a graphical representation of the integrals of the intensities of the Raman spectra in the regions related to CNTs and to polymers as a function of the planar position (X and Y) and focal depth (Z). The red (green) color is associated to the CNTs (polymers), and the higher the value of the integral in the region related to the CNTs (polymer), the brighter the red (green) color becomes.

Electrical resistance–strain data were collected by stretching, via a micrometric slide, conductive tracks made of the different composites, vertically attached to a 120 g metal weight placed on the plate of an analytical balance having 0.1 mg resolution. We then recorded simultaneously the resistance of the conductor and the variation in the gravitational force on the balance. Specimens were modeled in a “dog bone” shape (with construction proportion subject to the conventional ISO 527-1—“Plastics – Determination of the tensile characteristics”). In the case of our stretchable conductive composites, this experimental arrangement also allows for precise simultaneous measurement of the electrical resistance as a function of the strain in the same specimen. Specifically engineered copper clamps provide the necessary electromechanical connection of the specimens. General principles of mechanical characterization were followed, as reported in [24]. Such experimental setup allows a comparison of the mechanical and electrical behavior of the conductive specimens subject to strain, as shown by the plots in Figure 4, relative to the two different representative behaviors of thermoplastic polymers and thermosetting elastomers.

## 4. Discussion

The experimental evidence supplied by the different techniques employed in this work extends the preliminary analysis [19] on the strong and stable self-grafting of an SWCNT onto polymeric films. The microscopic process of the SWCNT self-grafting onto polymeric substrates is a complex process, strongly affected by fabrication methodologies. The body of results obtained here on composites prepared according to a uniform protocol shows, however, that grafting is very different between the class of thermoplastic substrates and that of elastomers.

More specifically, both the confocal Raman microscopy and SEM data show that the nanotube film grafted on the thermoplastic polymers is compact and thick, penetrating the polymer film up to approximately 20 μm (15 μm in the LDPE and up to 25 μm in the PP). Such depth is sufficient to withstand the brushing procedure devised to remove all the weakly bound CNT flakes. The organic/inorganic materials interact so that many nanotube bundles penetrate the polymer film for several microns, but also the polymer seems to embed the deposited SWCNT layer, contributing to binding it firmly to the substrate and stabilizing the whole layer.

Differently from what was observed with the thermoplastic polymers, in the case of the thermosetting elastomers, the SWCNT layer deposited on the film surface is not soaked by the polymer because of the very high polymer viscosity, which does not decrease significantly during the heating phase of the fabrication. The superficial layer is then easily and almost completely removed by the mechanical brushing. However, both the SEM images and Raman 3D maps show that the nanotubes penetrate deeply (between 70 and 100 µm) into the elastomeric substrates.

This different behavior might be ascribed to the lack of a semi-crystalline phase in strongly cross-linked elastomers. The consequent disordered structure and the density fluctuations, typical of the investigated thermosetting polymers, allow the easier penetration of nanotubes into the polymer structure, even if their viscosity is much higher than for thermoplastic polymers. On the other hand, the lower viscosity of thermoplastics may let these polymer chains drift, by capillary forces, into the layer of the deposited nanotube bundles; these become soaked and stabilized by such a “glueing” effect, which cannot occur in thermosetting elastomers.

The electrical characteristics of the polymers chosen as representative of the two classes (LDPE and PDMS) are strongly dependent on the discussed grafting mechanisms. The thick layer of the nanotube bundles remaining at the surface of the thermoplastic substrates yields a low electrical resistance, but given the poor elastic behavior of the thermoplastic substrate, it tends to crack at relatively low strains (<20%), thus rapidly and almost irreversibly increasing the conductor resistance. On the other hand, the more uniform dispersion of CNTs into elastomeric substrates causes a higher overall resistivity of the composite. As already noted, while for both types of polymer substrates the electrical behavior reported in Figure 4 seems driven by the corresponding mechanical behavior, the hysteresis is much larger for the thermoplastic substrate, both in stress and in resistance, most likely because of the slower relaxation of the LDPE polymeric chains with respect to the more cross-linked chains of the elastomeric substrate.

It is however interesting to observe in Figure 4 that the initial relaxation of the elastomer, unlike that of the thermoplastic LDPE, leads to a drop in the resistance. This behavior may be related to an increase in the alignment of the CNT bundles driven by the extension of the film, which brings a larger number of CNTs in contact with each other.

Both types of the investigated self-grafted composite types display resistivities in easily accessible ranges. As an example, the resistances of typical conductive tracks, 5 mm wide and 25 mm long, range from a few hundred Ω for thermoplastic substrates to a few MΩ for highly stretched (200–300%) elastomers, where the sensitivity is somewhat higher, reaching values of the gauge factor ΔR/R0 of the order of 5–10. Such a figure is still a factor 10${}^{2}$ lower than what is reported in the recent literature [8,9], thus limiting their application in the strain sensors field. However, two further needs, in the technology of stretchable conductive composites, are versatility and the ease of engineering [25], characteristics associated with the samples here described. The self-grafting method and the high conductance and elasticity of the as-grown bundled SWCNT ink make the design and development of a variety of different devices extremely easy and inexpensive. The further detailed electromechanical characterization of the investigated self-assembled composites is significant in light of the possible application in several fields all requiring elastic, stretchable and moldable electrical conductors. It is foreseeable that the conductive properties of different types of polymers could be exploited in peculiar environments, such as strongly vibrating systems, or in sensing devices. Moreover, although the systems could be used in individual microsystems, a scale-up toward the mass production of more complex devices is feasible by transferring the nanotube ”ink" from stamps rather than drop casting it onto stencils. Some type of particular CNT-polymer electrode arrays for electro-corticographic recordings have already proved to be very successful due to the excellent biocompatibility of polyethylene films and the adaptation of the arrays to the brain shape, and we envisage many other sensing and charge transfer application fields.

## 5. Conclusions

The interfaces of self-assembled SWCNT/polymer composite regions on polymeric films, both on the surface and in the bulk, have been investigated by several non-destructive techniques.

We have shown that the different grafting behaviors, and the consequent electro-mechanical responses, follow from the different nature of the polymer (i.e., thermoplastic or thermosetting), possibly because of both the cross-linking of the chains and from the presence or absence of a crystalline phase.

Such differences have important consequences on the electrical behavior of the stretchable conductors. The thick CNT layer, self-grafted on thermoplastic materials, yields a higher conductivity but also a non-constant conductance as a function of the strain, which drops very steeply above a strain of approximately 0.10–0.15, causing the permanent disconnection of the CNT bundles’ structures. This is reflected on the high hysteresis on the resistance–strain plot. This limits the application of this type of composite to low-strain devices.

On the other hand, elastomer-based CNT composites have a resistivity about two orders of magnitude higher and penetrate much more deeply into the polymer bulk. These conductors have a more linear behavior of the resistance–strain plot and can be stretched up to three times the original length, returning to the original resistance value with minimal hysteresis. They are thus more suitable for high-strain, heavy-duty devices. Our experimental investigation has thus provided new information on the penetration depth and surface grafting of an SWCNT on different kinds of polymers, together with a better understanding of how the macroscopic mechanical and electrical behavior depend on such grafting mechanisms.

These new findings will contribute to developing more and better performing CNT/polymer composites to improve our stretchable and moldable conductive devices and to associate them to the most appropriate application fields, particularly in the biomedical area, for example, in the technologies associated with prosthetics.

Further studies focused on a more thorough comprehension of the complex microscopic processes of grafting of a CNT on polymeric substrates are currently in progress.

## Figures and Tables

**Figure 1 molecules-28-01764-f001:**
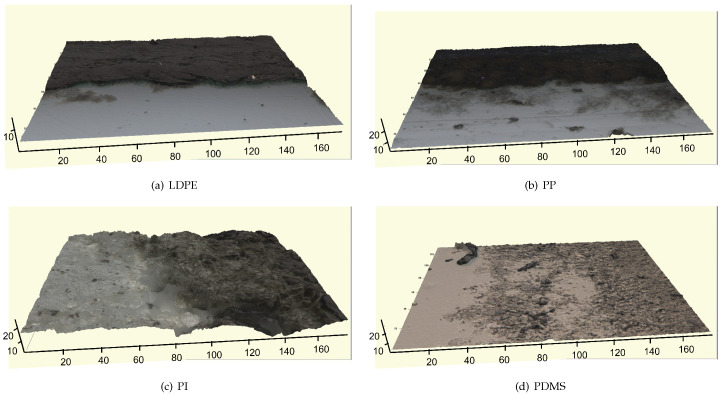
Surface optical profilometry images (all acquired with a 50 × objective) of conductive SWCNT tracks deposited onto different polymer substrates: (**a**) low-density polyethylene (LDPE), (**b**) polypropylene (PP), (**c**) polyisoprene (PI) and (**d**) polydimethylsiloxane (PDMS).

**Figure 2 molecules-28-01764-f002:**
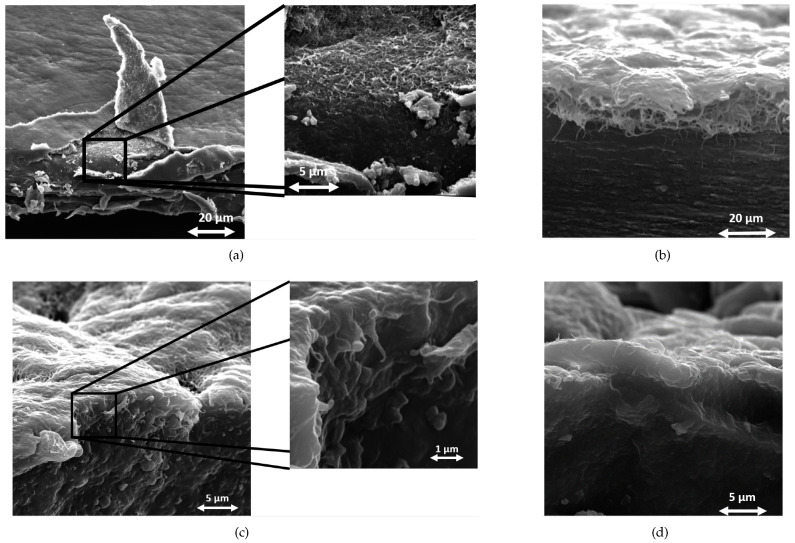
SEM images of SWCNT/polymer composite sections at low angle of incidence. (**a**) LDPE; (**b**) PP; (**c**) PI; (**d**) PDMS.

**Figure 3 molecules-28-01764-f003:**
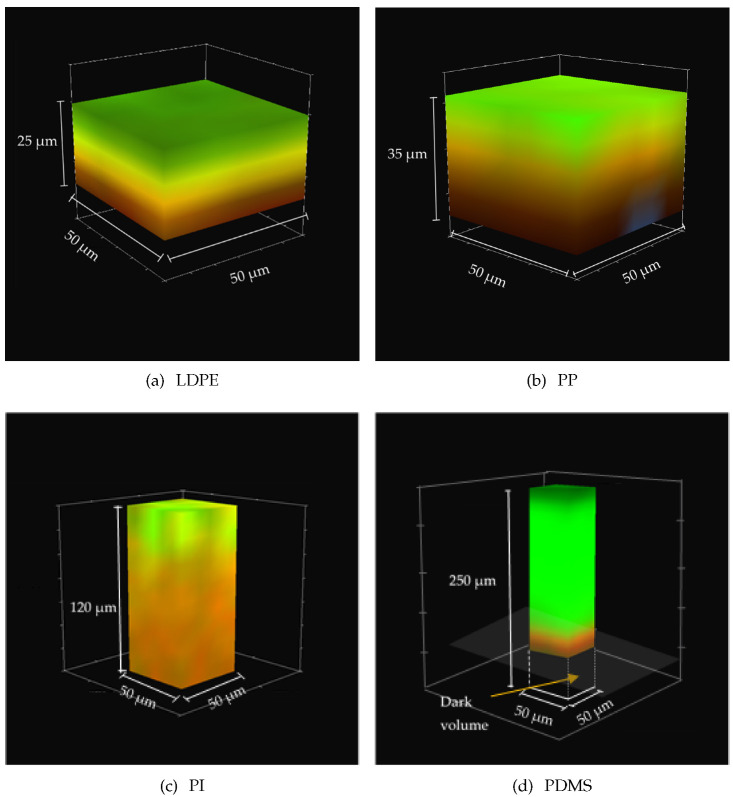
SWCNT penetration obtained from Raman spectral signature of carbon nanotubes (red) and of each polymer (green): (**a**) LDPE; (**b**) PP; (**c**) PI; (**d**) PDMS. Note that the laser beam impinges on the upper side of the 3D maps. In the case of the very thick PDMS substrate (**d**), the volume described as “dark volume” is the region that is not reached by the laser beam due to absorption by the SWCNT gradient.

**Figure 4 molecules-28-01764-f004:**
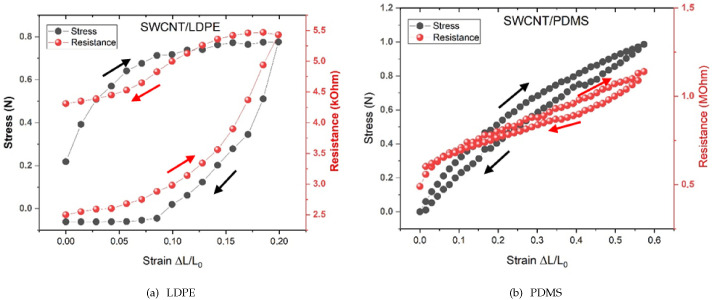
Mechanical and electrical behavior of self-assembled SWCNT/polymer composite films. Stress–strain and electrical resistance–strain plots for (**a**) thermoplastic (LDPE) and (**b**) thermosetting elastomeric (PDMS) polymer substrates. Resistance versus strain (in red) and stress versus strain (in black) are reported. Note the different y scales and the different shapes of cycles.

**Table 1 molecules-28-01764-t001:** RMS roughness of clean polymer substrates and SWCNT tracks and their step heights from optical profilometry. Measurements of nanotube depositions are taken after manual brush polishing.

Polymers	RMS Polymer (μm)	RMS SWCNT Track (μm)	Average Step Height (μm)
LDPE	0.2	0.3	3.0
PP	0.1	0.3	2.0
PI	1.7	3.3	−2.0
PDMS	0.1	0.6	0.7

## Data Availability

Not applicable.

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
