# Peer review of "Characterization of Conductive Carbon Nanotubes/Polymer Composites for Stretchable Sensors and Transducers"

_molecules, 2023, doi:10.3390/molecules28041764_

Round 1
Reviewer 1 Report
the manuscript is reviewed.
the comments are as follows-
title of the manuscript needs to modify.
author should include some quantitative data in the abstract, it will help to understand the manuscript better way.
author should provide in brief how the SWCNT is deposited into the film.
details about the specification of the instruments need to mention. eg. model no, origin etc.
I feel that sentence in line no 84 is not complete.
what are the magnifications of the images in figure 1 ?
What are the color represented in fig 3 ?
Did the author confirm the chemical composition of the materials?
what kind of instrument is used for mechanical properties ?
also the hydrophobic or hydrophilic properties are important to study.
author should provide a perspective of the study.
Author Response
We thank you for the suggestions, following comments and adding to the paper were reported point by point.
the manuscript is reviewed.
the comments are as follows-
- title of the manuscript needs to modify.
Accepting the referee's suggestion, we changed the title to:
Characterization of conductive carbon nanotubes/polymer composites for stretchable sensors and transducers
- author should include some quantitative data in the abstract, it will help to understand the manuscript better way.
The resistance range value for both groups of polymers that allows differentiation in the application of devices is now reported.
- author should provide in brief how the SWCNT is deposited into the film.
The text has been extensively modified adding details on the deposition procedure. The following text has been added:
“Purified SWCNT, produced by the HiPCo method [22] were purchased from Carbon Nanotech Technologies Inc. The diameter of the individual SWCNT is 0.7 nm, with both armchair and zig-zag species present. SWCNT were grafted on different polymer films by the drop-casting technique. A liquid vector solution was prepared by adding 0.1 mg of Linear Alkylbenzene Sulphonate (LAS) to 10 mg of 10:90 wt.% water/ethanol mixture. 1 mg of SWCNT was then added and dispersed in this solution by the combined surfactant action of LAS and of high power sonication by means of a Fisherbrand model 120 ultrasonic disintegrator for 60 minutes. Even after such treatment the SWCNT dispersion was never completely uniform, but keeping it under continuous sonication in a Elmasonic P ultrasonic bath operated at 80kHz, its homogeneity was high enough to cross through a 200 µm needle for drop casting purpose, by continuous sonication in a Elmasonic P ultrasonic bath operated at 80kHz. The vector fluid carrying the SWCNT dispersion was normally deposited by drop casting on suitable stencils designed to define the shape of the composite electrical conductors; to achieve them on the polymeric film substrates. The polymeric film substrates supported on a metal plate were heated to a controlled temperature. While the temperature was kept at approximately 80 °C, the CNT suspension was slowly cast through the needle on the stencil. Evaporation of the water/ethanol solution left a brown/black deposition the polymer surface. The operation was repeated until the space between the stencils were completely filled and appeared uniformly black. Irrespective of the drophobicity of the polymer surface, the vector droplets drying on the hot polymer surface cause some inevitable reaggregation of nanotube bundles into small flakes of random geometries. Each CNT track was therefore lightly brushed to remove non adhering flakes and further cleaned by a jet of purified air.To stabilize the deposition, the temperature was raised to a value only slightly lower than the thermoplastic polymer melting temperature (e.g. 115 °C for LDPE films, 150 °C for PP). Heating to 150 °C was also used for thermosetting elastomers (PDMS and PI). After heating for 1 minute, the samples were cooled to RT and mechanical brushing repeated. Finally, the stencil was carefully removed.
- details about the specification of the instruments need to mention. eg. model no, origin etc.
All the specifications of the instruments, have been added, and specification about ultrasonic disintegrator are now reported.
- I feel that sentence in line no 84 is not complete.
Thank you for your comment. In fact, the sentence was not complete due to a compilation error. Due to this mistake, two sentences were missing. We have added both sentences in the revised manuscript both missing sentences are present: the first describes the Raman analyses, the latter how the electro-mechanical measurements were performed.
- what are the magnifications of the images in figure 1 ?
- The magnification was reported only in the text, for the sake of clarity we have now added in the caption of Figure 1 that all the images have been obtained with a 50x objective.
- What are the color represented in fig 3 ?
Thank you for your comments, the explanation of this topic was probably not sufficient in the submitted version. In the revised version, we have added details about how the Raman 3D maps have been obtained, relating the colors to the intensities of the Raman peaks. Additional explanation is now reported in both the “Materials and Methods” and the “Results” sections and in Appendix A2.
- Did the author confirm the chemical composition of the materials?
The chemical compositions of the materials, given by the suppliers specifications, were confirmed by their Raman characteristic peaks and by Infrared Spectroscopy. This is now stated in the Materials and Methods section of the revised manuscript.
- what kind of instrument is used for mechanical properties ?
The apparatus is now described in the text:
“Electrical resistance-strain data were collected by stretching, via a micrometric slide, conductive tracks made of the different composites, vertically attached to a 120 g metal weight placed on the plate of an analytical balance having 0.1 mg resolution; we then recorded simultaneously the resistance of the conductor and the variation in the gravitational force on the balance. Specimens were modeled in a “dog bone” shape (with construction proportion subject to the conventional ISO 527-1 – “Plastics – Determination of the tensile characteristics. In the case of our stretchable conductive composites), this experimental arrangement also allows for precise simultaneous measurement of the electrical resistance as a function of the composite strain, thus relating instantaneously the mechanical and electrical behaviour of the same specimen. Specifically engineered copper clamps provide the necessary electromechanical connection of the specimens. General principles of mechanical characterization were followed reported in [ 25 ]). Such experimental set-up allows us to compare the mechanical and electrical behavior of the conductive specimens subject to strain as shown by the plots in Figure 4 relative to two different representative behaviors for thermoplastic polymers and thermosetting elastomers.
- also the hydrophobic or hydrophilic properties are important to study.
As now stated in the description of the samples fabrication, no evident effect of systems hydrophobicity has been observed, given the very rapid evaporation and the very small volume of the carrier fluid droplets.
- author should provide a perspective of the study.
The following text has been added:
“Further detailed electromechanical characterization of the investigated self-assembled composites is significant in light of the possible application in several fields all requiring elastic, stretchable and moldable electrical conductors. It is foreseeable that the conductive properties of different types of polymers could be exploited in peculiar environments such as strongly vibrating systems, or in sensing devices. Moreover, although the systems could be used in individual microsystems, a scale-up towards mass production of more complex devices is feasible by transferring the nanotube “ink” from stamps rather than drop casting it onto stencils.
Reviewer 2 Report
The paper “Carbon nanotubes/polymer composites for stretchable electrical sensors and transducers” reports the experimental studies of the interaction of carbon nanotubes deposited on two different groups of polymer film substrates, thermoplastic and thermosetting elastomers. The properties of CNTs-polymer composite films were studied by using different experimental techniques, profilometry investigation, SEM and Confocal Raman Microscopy. It has been reported different grafting behavior of two groups of composites based on thermoplastic and thermosetting elastomers. It was also shown how the macroscopic mechanical and electrical behavior depend on grafting mechanisms onto polymers surface. The obtained results are important taking into account an application potential.
In summary, this is experimentally interesting paper and, I can recommend the paper to be published in Molecules after minor revision.
Some remarks:
1. The introduction can be slightly modified incorporating the current research achievement in developing stretchable electrical sensors and highlighting the significance of the presented results.
2. The experimental part should be further developed; especially details on the determination of mechanical and electrical characterization and description of measurements uncertainties e.g. thickness of polymer films.
3. I suggest extended the discussion on the information related to the properties of CNTs-polymer composites available in the literature (it would be interesting to elaborate of the issue related to the influence of the fabrication methodologies on CNTs self-grafting process onto polymeric substrates).
4. In my opinion, it would be better to emphasize in the text that the SEM image from Appendix A1 concerns a different composite than the composites analyzed in the work.
5. Minor language/style corrections are suggested e.g. “allowing to successfully probe successfully their electro-corticographic signals” (line 33); I suggest use CNTs instead of CNT, e.g. "CNTs grafting onto the polymer" etc.
Author Response
We thank you for the suggestions, following comments and adding to the paper were reported point by point.
Some remarks:
- The introduction can be slightly modified incorporating the current research achievement in developing stretchable electrical sensors and highlighting the significance of the presented results.
Thank you for the comment, we have added in the introduction and in the discussion some recent references, and a brief description of sensor characteristics compared with significant results in the literature.
ACS Applied Materials & Interfaces. doi:10.1021/acsami.9b13684 (2019).
Tas, M., Baker, M. A., Masteghin, M. G., Bentz, J., Boxshall, K., & Stolojan, V. .Highly Stretchable, Directionally-oriented Carbon Nanotube/PDMS Conductive Films with Enhanced Sensitivity as Wearable Strain Sensors.
ACS Appl. Mater. Interfaces 2021, 13, 15572−15583
Wearable Strain Sensors Based on a Porous Polydimethylsiloxane Hybrid with Carbon Nanotubes and Graphene
ACS Appl. Mater. Interfaces 2020, 12, 17, 19874–19881
Facile Fabrication of High-Performance Pen Ink-Decorated Textile Strain Sensors for Human Motion Detection
- Cheng et al.
- The experimental part should be further developed; especially details on the determination of mechanical and electrical characterization and description of measurements uncertainties e.g. thickness of polymer films.
Details on the experimental procedures have been added (see points 3 and 9 in the response to Reewer 1).
- I suggest extended the discussion on the information related to the properties of CNTs-polymer composites available in the literature (it would be interesting to elaborate of the issue related to the influence of the fabrication methodologies on CNTs self-grafting process onto polymeric substrates).
Both types of the investigated self-grafted composites types display resistivities in easily accessible ranges. As an example, the resistances of typical conductive tracks, 5 mm wide and 25 mm long, range from a few hundred Ω for thermoplastic substrates to a few MΩ for highly stretched (200-300%) elastomers, where the sensitivity is somewhat higher, reaching values of the gauge factor ∆R/R0 of the order of 5-10. Such figure is still a factor 102 lower than what reported in the recent literature [tas2019- acsami.0c22823] thus limiting their application in the strain sensors field. However, one further need, in the technology of stretchable conductive composites, is versatility and ease of engineering [acsami.9b22534-], characteristics associated with the samples here described.
4. In my opinion, it would be better to emphasize in the text that the SEM image from Appendix A1 concerns a different composite than the composites analyzed in the work.
Thank you for your comment. There was a mistake in the caption, that wrongly reported as “MDPE” instead of “LDPE”. Actually, the SEM image is again from a CNT/LDPE composite, it is just a replica of the one sample reported in Figure 2.
5. Minor language/style corrections are suggested e.g. “allowing to successfully probe successfully their electro-corticographic signals” (line 33); I suggest use CNTs instead of CNT, e.g. "CNTs grafting onto the polymer" etc.
We agree with the reviewer in hence in the nomenclature should be consistent throughout the manuscript. However, CNT (or SWCNT) refers to the general carbon material which is made of individual nanotubes but also lumps, coils, ropes and bundles thereof, both of armchair and zig-zag type, while CNTs (or SWCNTs) refer to many individual single nanotubes. We have thus used the former acronym when dealing with the material because it is thje material that gives rise to interesting elastic and intertwining properties that individual nanotubes do not display.
Round 2
Reviewer 1 Report
the author addressed all the comments.
Author Response
Dear Reviewer
Thank you for your suggestions, we have improved the paper as indicated.
With respect to the version uploaded in 15/01/2023, we have accepted the reviewers’ suggestion to improve the quality of the paper.
In particular:
- the references have been checked, in particular taking into account the most relevant recent papers on similar composites;
- figures are now more clearly legible.
- other minor improvements and corrections have been made throughout the text.
- All changes are been highlighted writing in red
As far as the notation CNT vs CNTs is concerned, this is a controversial nomenclature issue, that arose in the early CNT days. We believe that an agreement is somehow now commonly accepted, to refer to CNT (and MWCNT or SWCNT) as a general material, while CNTs refer to more than one individual carbon nanotubes. Therefore it is common to refer to CNTs in simulation, or in Scanning Tunnelling Microscopy or in nanoelectronics work, where only few individual tubes are analyzed, and to CNT where CNT powder is used, composed of ropes, coils and bundles of millions of intertwined tubes, as a structural material. This use is analogous to that of carbon fiber: e.g., “many technological components are nowadays made of carbon fiber” (not of carbon fibers).
We hope that these changes have substantially raised the paper’s quality, and we thank the reviewers for their useful comments.
23/01/2023
Laura Fazi